# Hospital-Acquired Functional Decline and Clinical Outcomes in Older Cardiac Surgical Patients: A Multicenter Prospective Cohort Study

**DOI:** 10.3390/jcm11030640

**Published:** 2022-01-27

**Authors:** Tomoyuki Morisawa, Masakazu Saitoh, Shota Otsuka, Go Takamura, Masayuki Tahara, Yusuke Ochi, Yo Takahashi, Kentaro Iwata, Keisuke Oura, Koji Sakurada, Tetsuya Takahashi

**Affiliations:** 1Faculty of Health Science, Juntendo University, 3-2-12 Hongo, Bunkyo-ku, Ochanomizu Center Building 5F, Tokyo 113-0033, Japan; m.saito.tl@juntendo.ac.jp (M.S.); te-takahashi@juntendo.ac.jp (T.T.); 2Department of Rehabilitation, The Sakakibara Heart Institute of Okayama, 2-5-1 Nakai-Cho, Kita-ku, Okayama 700-0804, Japan; qqqz3sm9k@gmail.com; 3Department of Rehabilitation, Tsuchiya General Hospital, 3-30 Nakajima-Cho, Hiroshima 730-8655, Japan; reha-pt@tsuchiya-hp.jp; 4Department of Physical Therapy, Higashi Takarazuka Satoh Hospital, 2-1 Nagao-Cho, Takarazuka-shi 665-0873, Japan; m-tahara@mail.hts-hsp.com; 5Department of Rehabilitation, Fukuyama Cardiovascular Hospital, 2-39 Midori-Cho, Fukuyama-shi 720-0804, Japan; fch_reha@yahoo.co.jp; 6Department of Rehabilitation, Yuuai Medical Center, 50-5 Azayone, Tomishiro, Okinawa 901-0224, Japan; yo.takahashi7448@gmail.com; 7Department of Rehabilitation, Kobe City Medical Center General Hospital, 1-1-2 Minatojimaminami-Cho, Chuo-ku, Kobe-shi 650-0047, Japan; iwaken@kcho.jp; 8Department of Rehabilitation, Nozomi Heart Clinic, 3-5-36 Miyahara, Yodogawa-ku, Osaka-shi 532-0003, Japan; oura30155@gmail.com; 9Department of Rehabilitation, The Cardiovascular Institute, 3-2-19 Nishiazabu, Minato-ku, Tokyo 106-0031, Japan; sakura282517@gmail.com

**Keywords:** outcome assessment, functional decline, cardiovascular disease, cardiac surgery

## Abstract

This study aimed to determine the effect of hospital-acquired functional decline (HAFD) on prognosis, 1-year post-hospital discharge, of older patients who had undergone cardiac surgery in seven Japanese hospitals between June 2017 and June 2018. This multicenter prospective cohort study involved 247 patients with cardiac disease aged ≥65 years. HAFD was defined as a decrease in the short physical performance battery at hospital discharge compared with before surgery. Primary outcomes included a composite outcome of frailty severity, total mortality, and cardiovascular readmission 1-year post-hospital discharge. Secondary outcomes were changes in the total score and sub-item scores in the Ki-hon Checklist (KCL), assessed pre- and 1-year postoperatively. Poor prognostic outcomes were observed in 33% of patients, and multivariate analysis identified HAFD (odds ratio [OR] 3.43, 95% confidence interval [CI] 1.75–6.72, *p* < 0.001) and low preoperative gait speed (OR 2.47, 95% CI 1.18–5.17, *p* = 0.016) as independent predictors of poor prognosis. Patients with HAFD had significantly worse total KCL scores and subscale scores for instrumental activities of daily living, mobility, oral function, and depression at 1-year post-hospital discharge. HAFD is a powerful predictor of prognosis in older patients who have undergone cardiac surgery.

## 1. Introduction

It is important to assess the physical function of older cardiac surgical patients before surgery because poor physical function, which includes preoperative gait speed [1,2,3], frailty [4,5], and sarcopenia [6], is an independent poor prognostic factor. In particular, the gait speed is a simple and powerful assessment of physical function in older adults and is also used as a diagnostic criterion for frailty and sarcopenia [7,8,9]. In fact, previous studies have demonstrated that there is an association between gait speed and poor short-term prognosis in patients undergoing coronary artery bypass grafting or valvular surgery, suggesting that it is a crucial assessment tool in predicting the prognosis of older cardiac surgical patients [2,3].

Recently, hospital-acquired functional decline (HAFD) has garnered attention as a novel predictor of poor prognosis for hospitalized older patients. HAFD, which refers to the functional decline that develops in at least 20–40% of hospitalized older patients, can either be newly developed or a pre-existent condition that worsened during hospitalization [10,11,12,13]. HAFD is assessed by whether pre-hospital activities of daily living (ADL), instrumental ADL (IADL), or physical function have recovered at discharge, and it is reported to be related to in-hospital mobility and nutritional intake [11]. HAFD is a powerful poor prognostic predictor for hospitalized older patients [12,13,14,15], and the occurrence of HAFD in older cardiac surgical patients may be a prognostic predictor independent of preoperative low gait speed. However, the incidence of HAFD in older cardiac surgical patients, and the effect of HAFD occurrence on prognosis, are unclear.

Therefore, the purpose of this study was to determine the frequency of HAFD in older cardiac surgical patients and to examine whether the occurrence of HAFD is associated with a composite poor prognosis (severity of frailty, death, and cardiovascular readmission) one year after discharge.

## 2. Materials and Methods

This was a multicenter, prospective cohort study. A total of 281 patients with heart disease, aged ≥65 years, underwent elective cardiac surgery (coronary artery bypass graft, valvular disease surgery, or combined surgery) in seven Japanese hospitals between June 2017 and June 2018. The following exclusion criteria were applied: (1) a diagnosis of dementia; (2) an inability to walk independently or having bed rest due to severe preoperative heart failure; (3) in-hospital death; (4) data loss; (5) missing follow-up data.

### 2.1. Progression of Postoperative Rehabilitation

All patients started rehabilitation, under the guidance of a physiotherapist, the day after surgery. The postoperative rehabilitation protocol used for this study followed the Japanese Circulation Society Guidelines for the Rehabilitation of Patients with Cardiovascular Disease [16]. The rehabilitation started with active and passive movements in bed, with the ADL being extended gradually to sitting on the edge of the bed, standing, walking, aerobic exercise, and resistance training. Rehabilitation was performed five times per week for 60 min/day until the day before discharge.

### 2.2. Clinical Outcomes

The primary outcome included the composite outcomes of the severity of frailty, death, and cardiovascular readmission one year after hospital discharge. The severity of frailty is defined as a progression in the frailty category, during the one year after hospital discharge, compared with the preoperative status. The severity of frailty was assessed using the Kihon Checklist [17]. The KCL is a questionnaire that consists of 25 questions that can be answered with a yes/no. Overall scores can be stratified into three levels: robust (0–3 points), pre-frail (4–7 points), and frail (≥8 points) [18].

The secondary outcome was the change in the total scores and the scores of the seven domains of the KCL, administered preoperatively and one year postoperatively, in both groups (HAFD group vs. non-HAFD group). The 25 questions of the KCL are categorized into seven domains: IADL, mobility, nutrition, oral function, social, cognitive, and depression, enabling the analysis for each domain. This was important, as it allowed the problematic domains to be identified.

### 2.3. Definition of HAFD

HAFD was defined as a decrease in at least one point on the short physical performance battery (SPPB) before discharge compared to the score obtained before cardiac surgery [12,19]. The SPPB is a highly standardized geriatric physical functioning test that consists of assessments for balance, gait, strength, and endurance [20], and it is the highest recommended index in terms of validity, reliability, and responsiveness among the various physical function assessments used clinically in older adults [21]. Guralnik et al. reported that a 1-point change in the SPPB score results in a meaningful difference in mortality and risk of nursing home admission [20], with a minimal clinically important difference of one point [22]. Since the minimal clinically important difference in older cardiac patients who undergo rehabilitation during the hospitalization period is approximately one point, the HAFD in this study was defined as a decrease in the SPPB at discharge of at least one point from the preoperative level [23,24].

### 2.4. Clinical Characteristics and Measurements of Physical Function

The age, sex, body mass index (BMI), New York Heart Association cardiac function classification, comorbidity, and data from previous medical histories, as well as the results of investigations (left ventricular ejection fraction, hemoglobin, albumin, and estimated glomerular filtration ratio) were obtained from the medical records. All the preoperative clinical data were measured or obtained between the day before the surgery and the day of the surgery. Preoperative frailty was defined as a total KCL score of ≥8 points [18]. Data regarding surgical procedure, operation time, and intraoperative bleeding were collected from the surgical records. The postoperative course of the patients was recorded as the number of days spent in the intensive care unit, the postoperative day on which rehabilitation started, the postoperative day on which ambulation started and when ambulation independence was achieved, and the duration of the hospital stay.

The physical function was assessed using the SPPB, grip strength, and gait speed before surgery and at discharge. The SPPB was assessed using the SPPB manual [20]. The grip strength was measured with a Jamar hand grip dynamometer (Nihon Medix, Chiba, Japan), with the patients seated on a chair, their knees bent at 90° flexion, and the forearms in a neutral position. The gait speed was measured using a 4-m course, with the patients instructed to walk from the start to finish at their normal pace, while a stopwatch measured the time it took for them to finish the course. This test was performed twice, and the shortest time taken was used for the analysis.

The Asian Working Group for Sarcopenia specified the cut-off values for diagnosing sarcopenia as a grip strength of 26 kg for men, 18 kg for women and a gait speed of 0.8 m/s [8]. Preoperative grip strength and gait speeds below the cut-off values were defined as “low preoperative gait speed” and “low preoperative grip strength.”

### 2.5. One-Year Follow-Up Data

One year after discharge from the hospital, follow-up surveys were conducted by mail to determine patient survival, cardiovascular-related readmissions, and the KCL score.

#### Statistical Analysis

Continuous variables were expressed as median (interquartile range [IQR]), because they were not normally distributed, and categorical variables were expressed as number and percentage. The two groups (HAFD and non-HAFD groups) were compared using the chi-square test, for categorical covariates, or the Mann–Whitney U-test. A 2-sided *p*-value < 0.05 was considered statistically significant. Univariate and multivariate analyses were used to determine the odds ratio for each factor, to extract factors involved in the primary outcome of poor prognosis, one year after hospital discharge in an exploratory manner. To determine the influence of the relationship between the outcomes, variables with *p*-values <0.05 in the univariate analysis, and those deemed to be clinically important, were entered into a multivariate analysis. To avoid collinearity, the correlation coefficients between each parameter were determined and confirmed as not highly correlated. In a sub-analysis examining the interaction between HAFD and low preoperative gait speed, which increases the risk of poor prognosis, the patients were divided into four groups, according to HAFD and low preoperative gait speed, and logistic regression analysis was performed with poor prognosis as the dependent variable. A two-way analysis of variance was used for the secondary outcome and the change in preoperative and postoperative KCL scores between the two groups. All analyses were performed using IBM SPSS Statistics for Windows, Version 21.0 (IBM Corp., Armonk, NY, USA).

## 3. Results

### 3.1. Study Population and Incidence of HAFD

Among the 281 patients who were enrolled in the study initially, 34 patients were excluded, including 2 patients who died in-hospital, 10 patients whose data was lost, and 22 who had missing follow-up data. The baseline demographics and characteristics of the study population are shown in Table 1 and Table 2. By definition, 52 of 247 patients (21%) experienced HAFD after cardiac surgery.

The HAFD group had a significantly higher percentage of females, higher rates of chronic obstructive pulmonary disease (COPD), and lower preoperative hemoglobin levels, as well as grip strength, compared to the non-HAFD group. The HAFD group also had a significantly lower SPPB at discharge compared to the non-HAFD group.

### 3.2. Association between HAFD and the Primary Outcome

The primary outcome, poor prognosis, was observed in 82 patients (33%), severity of frailty in 57 patients (23%), death in four patients (2%), and cardiovascular-related rehospitalization in 21 patients (9%). After performing the univariate analysis, the age, sex, BMI, left ventricular ejection fraction, hemoglobin level, low preoperative gait speed, operative time, and HAFD were included in the multivariate regression analysis (Table 2). The results showed that HAFD (OR 3.437, 95% CI 1.756–6.729, *p* < 0.001), and low preoperative gait speed (OR 2.477, 95% CI 1.185–5.176, *p* = 0.016) were associated independently with poor prognosis.

Figure 1 shows the risk of poor prognosis for the interaction between low preoperative gait speed and HAFD. The combination of both low preoperative gait speed and HAFD (OR 12.84, 95% CI 2.61–63.08) showed a greater increase in the incidence of poor prognostic outcomes compared to low preoperative gait speed (OR 2.14, 95% CI 0.99–4.61) or HAFD (OR 3.21, 95% CI 1.59–6.50) alone.

### 3.3. Changes in the Kihon Checklist Score among the HAFD and Non-HAFD Groups

Figure 2 show the changes in the KCL score in the HAFD and non-HAFD groups before surgery and one year after hospital discharge. The two groups showed a significant main effect and an interaction between the two groups on the total KCL scores (F = 10.55, *p* < 0.001) and IADL (F = 4.29, *p* < 0.05), mobility (F = 10.44, *p* < 0.001), oral function (F = 7.27, *p* < 0.01), and depression (F = 6.11, *p* < 0.05).

## 4. Discussion

This study clarified the effect of HAFD on poor prognosis, one year after discharge, in older cardiac surgical patients. To the best of our knowledge, this is the first study to report on older cardiac surgical patients who have undergone standard open-heart surgery, although there have been previous studies on patients who have undergone minimally invasive transcatheter aortic valve implantation [12].

We found that the incidence of HAFD in older cardiac surgery patients was 21%. This was consistent with the findings of previous studies that reported that the incidence of HAFD was approximately 20–40% [10,11,12]. HAFD has also been shown to be related to in-hospital mobility, nutrition intake, and continence care, as well as to the length of hospital stay and the condition of the patient before hospitalization [11]. In this study, we found a significantly higher proportion of patients with COPD, significantly lower hemoglobin levels, and preoperative grip strength, as well as a higher proportion of females in the HAFD group compared to the non-HAFD group. It has been reported that patients with COPD, and those with a low preoperative forced expiratory volume in one second, had a prolonged duration of postoperative ventilator use, a higher incidence of postoperative respiratory complications, and in-hospital mortality [25,26]. Preoperative abnormalities in lung function, due to COPD, may delay the recovery of physical function after surgery. A recent meta-analysis of studies concluded that preoperative anemia was associated with poor outcomes after surgery [27]. Since preoperative anemia was associated with an increased amount of red blood cell transfusions [27], we speculated that the high degree of postoperative anemia was associated with a lower rate of physical inactivity and a higher incidence of HAFD. Although there was no significant difference in the progression of postoperative rehabilitation between the HAFD and non-HAFD groups, we speculated that the HAFD group tended to have a lower preoperative reserve capacity and did not fully recover their physical function at the time of discharge from the hospital due to the surgical invasion.

The total KCL score in the HAFD group, one year after discharge, was significantly higher than both the preoperative scores and that of the non-HAFD group, which was interpreted as an increase in the severity of frailty. The total KCL score of the HAFD group one year after discharge was 7.7 points, and considering that a total score of eight points or more corresponds to frailty [18], many in the HAFD group were likely to be in a frail state. A higher total KCL score has been associated with increased mortality and a higher risk of requiring long-term care insurance services [18,28,29,30]. In particular, increases in the severity of frailty stratification scores have been found to be associated with increases in the mortality rate [31] and in the rate of new forms of long-term care service and support required [18]. Recently, a large multicenter study reported that multifaceted frailty (physical/social/cognitive), in older patients with cardiac disease, increased the risk of readmission and death [32]. An increase in the severity of frailty has also been shown to substantially increase healthcare costs [33,34] with major effects on society, including further poor prognosis and more healthcare professionals required to care for these patients. Therefore, the selection of frailty severity as a clinical outcome in this study, in addition to death and rehospitalization, appears to be an important and appropriate outcome measure of poor prognosis. 

The KCL subtests of mobility, IADL, oral function, and depression were scored higher than the preoperative and non-HAFD groups. The occurrence of HAFD indicated that mobility had not recovered, even after 1 year of discharge, suggesting that the decline in motor function may have caused IADL and depression. In our study, 33% of patients had a poor prognosis. For example, Govers et al. reported that 38% of older cardiac surgical patients (aged 65–79 years) had decreased ADL scores one year after discharge [35], which was similar to previous studies.

HAFD was the most relevant predictor for poor prognosis one year after discharge. In previous studies, gait speed has been used, clinically, as an important prognostic predictor after cardiac surgery [1,2]. A previous multicenter study also reported that preoperative walking speed was an important predictor for postoperative functional recovery [36]. In the present study, preoperative gait speed was also identified as an independent predictor for poor prognosis in the multivariate analysis. However, in the present study, HAFD was found to be a more powerful predictor than the preoperative gait speed. This finding indicated that, even if the gait speed was normal preoperatively, the prognosis worsened when HAFD occurred postoperatively. Therefore, a prognostic prediction that considered the degree of recovery of physical function after surgery is important. Furthermore, considering that the prognosis of poor outcome is 12 times higher when low preoperative gait speed and HAFD occurred together (compared to no low preoperative gait speed or HAFD), the evaluation of HAFD is important in clinical practice.

In recent years, the advances in surgical techniques have expanded the scope of surgery to include older and severely ill patients, while acute care hospitals have shortened the length of hospital stays. Therefore, the number of patients with HAFD is expected to increase in the future, making the findings of this study significant. When considering surgical treatment for older patients, HAFD should be considered, and physical function should be monitored regularly by physiotherapists and nurses before and after surgery. Simple exercise (walking and chair stand) has been reported to reduce HAFD [37]. For patients with delayed recovery of postoperative physical function, the occurrence of HAFD may be prevented with an active improvement of physical activity and the incorporation of programs to increase the physical function in postoperative care.

### Limitations

First, the sample size was small. Moreover, a number of patients in each group did not respond to the post-discharge survey, which may have affected the post-discharge survey results. While the first author did not participate in data analysis, the co-authors participated in the measurements at each site, so the possibility that they had some influence on the results cannot be ruled out completely. The median preoperative SPPB of patients in this study was 12 points, and many had a high preoperative physical function. In addition, the KCL is a self-administered questionnaire, and patients with obvious dementia before surgery were excluded from the study. Therefore, the results of this study are biased toward older cardiac surgery patients whose physical and cognitive functions are relatively well preserved. In addition, since this study aimed to investigate the composite outcome one year post-hospital discharge, the speed of occurrence of the outcomes of death and readmission was not examined. Further studies are required to examine the timing concerning the occurrence of disability in future. Furthermore, this study did not address the cause of HAFD. In the future, it is necessary to examine factors, such as delirium, that lead to HAFD. Finally, this study was carried out in Japan. Thus, the results of this study may not be applicable to patients from other countries.

## 5. Conclusions

HAFD occurred in 21% of older cardiac surgical patients and was an independent predictor for poor prognosis one year postoperatively. More importantly, HAFD was a more powerful prognostic predictor than low preoperative gait speed, and the combination of low preoperative gait speed and the occurrence of HAFD increased the odds ratio 12-fold. Since HAFD was the most relevant prognostic predictor in older cardiac surgical patients, the prognosis should include both preoperative and postoperative functional recovery.

## Figures and Tables

**Figure 1 jcm-11-00640-f001:**
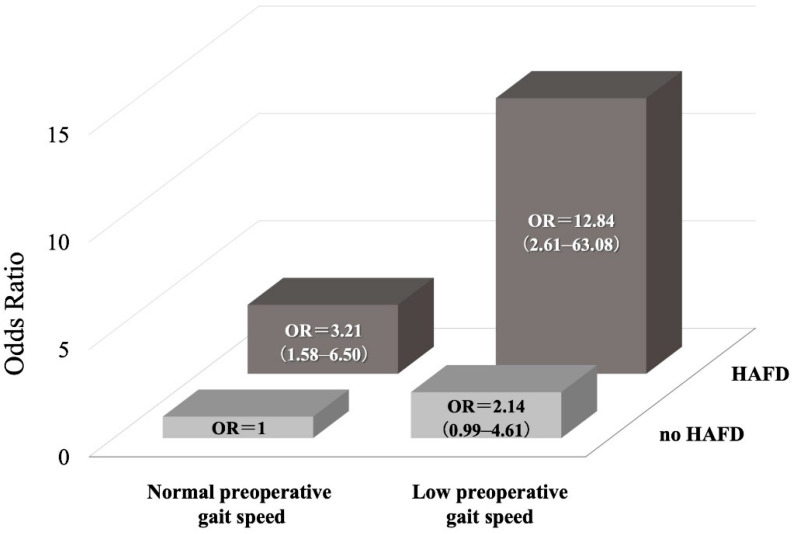
The interplay between low preoperative gait speed and HAFD increases the risk of poor prognosis. OR, odds ratio; HAFD, hospitalization-acquired functional decline.

**Figure 2 jcm-11-00640-f002:**
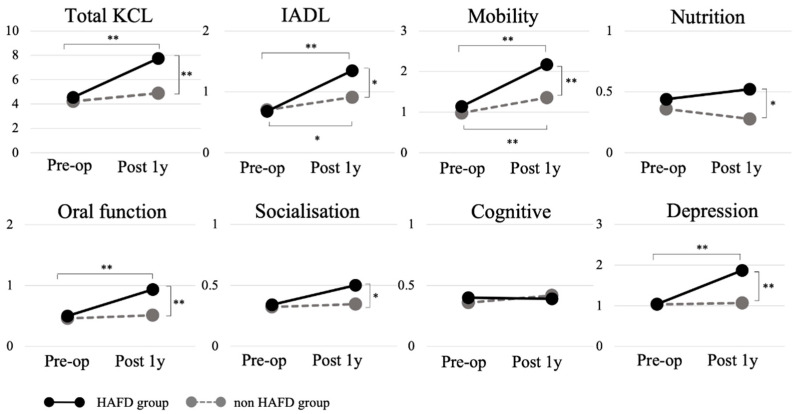
Changes in the Kihon Checklist score between the HAFD and non-HAFD groups. HAFD, hospitalization-acquired functional decline; KCL, Kihon Checklist; IADL, instrumental activities of daily living; Pre-op, preoperative; Post 1y, 1 year post hospital discharge. ** *p* < 0.01, * *p* < 0.05.

**Table 1 jcm-11-00640-t001:** Patient clinical characteristics.

	All (*n* = 247)	HAFD Group (*n* = 52)	Non-HAFD Group (*n* = 195)	*p*-Value
Age, years	74.0 (69, 79)	75.0 (69, 80)	75.0 (68, 80)	0.231
Sex, female, % (n)	38 (95)	50 (26)	35 (69)	0.040 *
Body mass index, kg/m^2^	23.1 (21.0, 25.3)	23.1 (19.8, 25.4)	23.6 (21.8, 25.6)	0.222
NYHA class, % (n)				0.568
Class I/ Class II/ Class III/ Class IV	38 (94)/52 (129)/9 (21)/1 (3)	44 (23)/46 (24)/10 (5)/0 (0)	37 (71)/54 (105)/8 (16)/1 (3)	
LVEF, %	63 (55, 70)	64 (55, 71)	64 (56, 71)	0.640
Comorbidity				
Diabetes mellitus, % (n)	34 (84)	44 (23)	31 (61)	0.058
Chronic kidney disease, % (n)	22 (55)	25 (13)	18 (35)	0.323
Chronic heart failure, % (n)	39 (95)	35 (18)	40 (77)	0.631
Chronic obstructive pulmonary disease, % (%(n)	6 (14)	14 (7)	4 (7)	0.013 *
Cerebrovascular disease, % (n)	15 (32)	14 (7)	13 (25)	0.530
Hemoglobin, g/dL	12.9 (11.6, 14.1)	12.5 (11.6, 13.6)	13.3 (11.7, 14.5)	0.025*
Albumin, g/dl	4.0 (3.7, 4.2)	4.1 (3.8, 4.2)	4.0 (3.7, 4.2)	0.787
eGFR, ml/min/1.73 m^2^	59.3 (44.3, 70.1)	56.0 (38.6, 65.9)	59.3 (45.7, 70.8)	0.117
Preoperative SPPB score, points	12 (10, 12)	12 (11, 12)	12 (11, 12)	0.199
Postoperative SPPB score, points	12 (10, 12)	10 (9, 11)	12 (11, 12)	<0.001 *
Preoperative gait speed, m/s	0.98 (0.83, 1.13)	0.97 (0.82, 1.04)	1.03 (0.88, 1.16)	0.152
Preoperative grip strength, kg	23.7 (17.9, 31.0)	20.2 (16.3, 26.7)	25.0 (18.5, 32.1)	0.002 *
Preoperative frailty, % (n)	25 (61)	29 (15)	24 (46)	0.470
Type of Operation, % (n)				0.441
CABG/Valve surgery/	26 (64)/32 (80)	27 (14)/31 (16)	26 (50)/33 (64)	
Multiple valve surgery/	23 (56)	17 (9)	24 (47)	
CABG + valve surgery	19 (47)	25 (13)	17 (34)	
Operation time, min	300 (251, 351)	288 (245, 332)	302 (243, 365)	0.691
Bleeding, mL	570 (320, 1218)	471 (320, 970)	610 (260, 1350)	0.732
Length of ICU stay, days	4.0 (3.0, 5.0)	4.0 (3.0, 5.0)	3.0 (2, 4)	0.142
Postoperative day that rehabilitation was started, days	1.0 (1.0, 1.0)	1.0 (1.0, 1.0)	1.0 (1, 1)	0.370
Postoperative day that ambulation was started, days	3.0 (2.0, 4.0)	3.0 (2.0, 4.0)	3.0 (2, 4)	0.229
Postoperative day when ambulation independence was achieved, days	5.0 (4.0, 6.3)	5.0 (5.0, 7.0)	5.0 (4, 7)	0.180
Length of hospital stay, days	19.0 (16.0, 25.0)	22.0 (15.0, 27.0)	19.0 (16, 24)	0.132

Note. HAFD, hospital-acquired functional decline; LVEF, left ventricular ejection fraction; NYHA, New York Heart Association; eGFR, estimated glomerular filtration ratio; SPPB, short physical performance battery; CABG, coronary arterial bypass graft; ICU, intensive care unit. Values are presented as median (interquartile range) or n (%). * *p* < 0.05.

**Table 2 jcm-11-00640-t002:** Predictors of all-cause mortality, readmission, and frailty severity, according to the univariate and multivariate regression analyses.

	Univariate Analysis				Multivariate Analysis			
	OR	95% CI		*p*-Value	OR	95% CI		*p*-Value
Age (every 1-year increase)	1.035	0.989	1.084	0.137	1.027	0.975	1.082	0.317
Female	1.153	0.670	1.986	0.607	1.173	0.609	2.256	0.634
BMI (every 1-kg/m^2^ increase)	0.966	0.895	1.043	0.378	1.002	0.917	1.095	0.964
NYHA class ≥ III (every degree increase)	1.529	0.648	3.610	0.333				
LVEF (every 1% increase)	0.977	0.957	0.998	0.031 *	0.982	0.959	1.005	0.129
Diabetes mellitus	1.220	0.700	2.127	0.483				
CKD	1.448	0.755	2.778	0.266				
Hemoglobin	0.805	0.687	0.942	0.007 *	0.847	0.701	1.023	0.085
Albumin	0.670	0.349	1.285	0.228				
Low preoperative gait speed	2.318	1.200	4.479	0.012 *	2.477	1.185	5.176	0.016 *
Low preoperative grip strength	1.046	0.598	1.828	0.875				
Preoperative SPPB score	0.937	0.816	1.077	0.361				
Bleeding	1.000	1.000	1.000	0.357				
Operative time	1.003	1.000	1.006	0.093	1.004	1.000	1.007	0.051
Postoperative ICU stay	1.120	0.980	1.282	0.097				
Postoperative hospital stay	0.994	0.967	1.023	0.690				
Hospital-acquired functional decline	3.467	1.842	6.528	<0.001 **	3.437	1.756	6.729	<0.001 **

Note. BMI, body mass index; NYHA, New York Heart Association; LVEF, left ventricular ejection fraction; CKD, chronic kidney disease; SPPB, short physical performance battery; ICU, intensive care unit; OR, odds ratio; CI, confidence interval. * *p* < 0.05; ** *p* < 0.001.

## Data Availability

The dataset(s) supporting the conclusions of this article cannot be provided due to ethical restrictions.

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
