# Peer review of "Hospital-Acquired Functional Decline and Clinical Outcomes in Older Cardiac Surgical Patients: A Multicenter Prospective Cohort Study"

_jcm, 2022, doi:10.3390/jcm11030640_

Round 1
Reviewer 1 Report
In my opinion this is a very well written and presented article. Authors from Japan have studied an important aspect of treatment and follow up in the elderly population. I would like to ask if it was possible to include duration of extracorporeal circulation to the tables as this is not always same number as surgery duration and seems to be important factor. Additionally history of stroke or TIA would be interesting in the tables.
Author Response
Dear Reviewer
We would like to thank you very much for reviewing our paper. We have received very important comments and have revised the paper.
Comments and Suggestions for Authors
In my opinion this is a very well written and presented article. Authors from Japan have studied an important aspect of treatment and follow up in the elderly population. I would like to ask if it was possible to include duration of extracorporeal circulation to the tables as this is not always same number as surgery duration and seems to be important factor. Additionally history of stroke or TIA would be interesting in the tables.
>Thank you for your very important point. We have added a history of cerebrovascular disease to Table 1. However, the extracorporeal circulation time could not be added to the additional information due to difficulties in collecting data from all of the sites. We recognize that this very important indicator should be included in future studies.
Reviewer 2 Report
The authors try to answer whether HAFD is associated with poor outcomes in older cardiac surgical patients one year after hospital discharge. However, the study has several major disadvantages:
- Abstract is unclear. It is not necessary to explain which statistical methods did you use in the abstract. On the other hand, the abstract should contain definitions of HAFD, primary and secondary outcomes. Also, it should be emphasized that this was multicenter prospective study in this section. The results with p values should be add to the accompanying sentences regarding independent predictors of primary and secondary outcomes at one-year follow-up.
- In the section „ Clinical Outcomes“, sentences from lines 101 till 111 should be placed in the section „Discussion“.
- Statistical analyzes are incorrect. Kaplan-Meier should be used to determine differences in primary outcomes at one-year follow-up between HAFD and non-HAFD groups. Also, Cox regression analyses should be used for univariate and multivariate analysis regarding primary outcomes. Therefore, it is better to define primary outcomes as the composite of death and cardiovascular readmission at one-year follow-up, while the severity of frailty should be the part of secondary outcomes. Then, univariate and multivariate logistic regression analysis should be used to determine predictors of secondary outcomes at one-year follow-up.
- Figure 1 should be presented as Kaplan-Meier curve rather than with bar graphs.
- In the section „ Changes in the Kihon Checklist Score among the HAFD and Non-HAFD groups“ it is not necessary to explain statistical analysis that you used.
- Table 3 is unnecessary.
- English language editing is advised.
Author Response
Dear Reviewer
We would like to thank you very much for reviewing our paper. We have received very important comments and have revised the paper.
1. Abstract is unclear. It is not necessary to explain which statistical methods did you use in the abstract. On the other hand, the abstract should contain definitions of HAFD, primary and secondary outcomes. Also, it should be emphasized that this was multicenter prospective study in this section. The results with p values should be add to the accompanying sentences regarding independent predictors of primary and secondary outcomes at one-year follow-up.
>Thank you for your very valuable comments. The abstract has been completely revised according to your suggestion.
2. In the section „ Clinical Outcomes“, sentences from lines 101 till 111 should be placed in the section „Discussion“.
>Thank you for your comment. As per your suggestion, we have moved some sentences from the “Clinical outcome” section to the “Discussion” section. Some sentences in the “Discussion” section have been revised accordingly.
3. Statistical analyzes are incorrect. Kaplan-Meier should be used to determine differences in primary outcomes at one-year follow-up between HAFD and non-HAFD groups. Also, Cox regression analyses should be used for univariate and multivariate analysis regarding primary outcomes. Therefore, it is better to define primary outcomes as the composite of death and cardiovascular readmission at one-year follow-up, while the severity of frailty should be the part of secondary outcomes. Then, univariate and multivariate logistic regression analysis should be used to determine predictors of secondary outcomes at one-year follow-up.
>Thank you for your very important comments. In this study, we were interested in determining the prognosis 1-year post-hospital discharge, based on previous studies. Therefore, univariate and multivariate analyses and odds ratios related to poor prognosis were determined. However, as you have noted, the speed of disability occurrence is also a very important issue; therefore, the points you have mentioned have been included in the “Limitations” section.
4. Figure 1 should be presented as Kaplan-Meier curve rather than with bar graphs.
>Thank you for your important suggestion. As noted earlier, the primary outcome of this study was to examine the composite outcome of death, rehospitalization, and progression of frailty severity at 1-year post-hospital discharge. We agree that the speed of failure is an important point, and we have added this to the “Limitations” section.
5. In the section „ Changes in the Kihon Checklist Score among the HAFD and Non-HAFD groups“ it is not necessary to explain statistical analysis that you used.
>Thank you. We have removed the relevant text concerning the statistical methods used.
6. Table 3 is unnecessary.
>Thank you for your comment. Table 3 has been deleted.
7. English language editing is advised.
>Thank you for your advice. This paper was edited by a professional editing service at the time of submission, and the modifications to the text have been edited prior to resubmission.

Round 2
Reviewer 2 Report
The revised manuscript is much better than first one. The manuscript is clearer, results are well reported and discussion is balanced. However, I have a few more concerns that need to be evaluated:
- In the Abstract, line 42, instead of the word „factor“, it is better to stand „predictors“, because HAFD and low preoperative gait speed are independent predictors (not factors) of poor prognosis.
- Percentages in the Abstract, text and in all Tables should be presented as the whole number. The same goes for „Age”.
- In the Table 1, all continuous variables should be presented as for example “Age + SD, years", or “Body mass index + SD, kg/m2”, etc.
Author Response
Dear Reviewer
Thank you very much for reviewing our manuscript. We thank you for your important comments. We have revised the manuscript according to very valuable comments.
1) In the Abstract, line 42, instead of the word „factor“, it is better to stand „predictors“, because HAFD and low preoperative gait speed are independent predictors (not factors) of poor prognosis.
>Thank you for your very valuable comments. We have changed the word "factor" to "predictor(s)" in the abstract and manuscript.
2) Percentages in the Abstract, text and in all Tables should be presented as the whole number. The same goes for „Age”.
>Thank you for your very valuable comments. All percentages and ages have been corrected to integers.
3) In the Table 1, all continuous variables should be presented as for example “Age + SD, years", or “Body mass index + SD, kg/m2”, etc.
>Thank you for your very valuable comments. Since the data in Table 1 are not normally distributed, there was a possibility that the data would be affected by outliers, so the median (quartile) was used. However, this was not mentioned in the "Statistical Analysis" section, so it was added in the manuscript.